# Expanding Workplace Inclusion of Employees Who Are Parents of Children with Disabilities through Diversity Training

**DOI:** 10.3390/healthcare10122361

**Published:** 2022-11-24

**Authors:** Lisa M. Stewart, Julie M. Rosenzweig, Anna M. Malsch Tamarkin, Eileen M. Brennan, Jessica Lukefahr

**Affiliations:** 1College of Health Sciences and Human Services, California State University, Monterey Bay, 100 Campus Center, Seaside, CA 93955, USA; 2School of Social Work, Portland State University, P.O. Box 751, Portland, OR 97207, USA; 3NPC Research, 975 SE Sandy Blvd., Portland, OR 97214, USA

**Keywords:** human resource development, diversity, inclusion, multiple-role management, work-family, exceptional caregiving

## Abstract

Employed parents raising children with disabilities manage exceptional care responsibilities along with their work careers. This study examines the effects of targeted diversity training on human resource (HR) professionals’ knowledge of the work–family experiences of these parents, and on their self-efficacy in providing workplace supports. Using computer-based training in field settings, 64 U.S. human resource professionals in an international company participated in two diversity training sessions. Data related to knowledge and efficacy of dependent and disability care were collected before the first training and immediately after the second. HR participants demonstrated significant increases from pretest to posttest on the trained items: knowledge of dependent and disability care and self-efficacy regarding provision of workplace supports. There was no change in relevant but untrained variables over time. Training HR professionals on parents’ exceptional care responsibilities, specific community resources, and heightened self-efficacy promoted their likelihood to grant flexible work arrangements. Results suggest HR self-efficacy is developmental, building on prior knowledge of dependent care and tenure in HR positions. This is one of the first studies that address the effects of HR diversity training regarding employees providing exceptional care. Theoretical developments and implications for inclusive practices are discussed.

## 1. Introduction

Organizations continue to expand their initiatives on diversity management by actively committing to an *inclusive workplace*, which may heighten employees’ organizational commitment, engagement, and retention [1,2]. Initiatives most often include diversity of sexual orientation, gender identity and expressions, religious practices, culture, ability, and age; however, the unique experiences and voices of employees caring for dependent children with disabilities remains an underdeveloped facet of workplace diversity and inclusion [3,4]. Given that approximately 9% of employees in any given company in the U.S. are caring for a child with a disability or chronic health condition [5], there is a substantial group of employees that remain hidden from these current initiatives. These employed parents engage in intensive management of work and family roles due to the demands of their parenting [6]. This paper reports on an intervention designed to increase the knowledge of human resource (HR) professionals regarding the role-management challenges facing these employees, as well as their self-efficacy in providing workplace supports to assist parents providing disability care.

### 1.1. The Exceptional Caregiving Experience

Parenting a child with a disability or chronic condition, including mental health disorders, is a type of dependent care known as *exceptional caregiving* [7,8]. Exceptional caregiving requires that families devote intense physical, emotional, and financial resources that can change over time due to developmental demands and/or changes in the chronicity of the disability [9]. In contrast to raising a child with typical developmental needs, caring for a child with a disability often brings more challenges and complexities across key developmental stages for both the child and the parent, frequently impacting the health and well-being of the caregiver [8,10].

It is important to acknowledge that employees may be facing complex family situations that include their own disabilities and those of their partners, parents, and one or more children or dependents for whom they provide or manage care. As the 2022 National Strategy to Support Family Caregivers indicates, a major goal in the United States is to ensure financial and workplace security for those engaged in caregiving as well as employment [11]. The 2019 American Community Survey found that 2.6 million families reported they cared for at least one child with a disability in their home, representing 7.2% of all households including children under 18 [12].

Employers are beginning to recognize that workplace interruptions may be greater for employees with children who have disabilities compared with employees parenting children with typical development. One international study on the global workforce found that employed parents with disability-related care responsibilities for their children were more likely to reduce their employment or withdraw from it compared to other groups of employees with family care responsibilities (typically developing child, older adult, disability care) [13]. In a recent study, Stefanidis and Strogilos [14] established that higher levels of support from supervisors and co-workers are associated with greater workplace engagement for employed parents of children with disabilities. With appropriate supports, such as a supervisor willing to allow for informal requests for flexibility, community resources such as medical support services, and inclusive after school programs and clubs, parents raising young people with special needs can thrive [15,16].

Since human resource (HR) professionals are key organizational leaders that assist supervisors, managers, and employees to handle workplace challenges, the initiative reported here focused on strengthening their knowledge of the issues faced by this substantial group of employees and their confidence to engage in supportive practices. Key elements of the training initiative included supportive practices such as advocating for workplace flexibility, improving the workplace culture, and engaging in diversity training. It also considered how HR professionals can help employees access resources through supportive organizational policies such as workplace flexibility and link employees with resources that are available in their communities, such as parent support groups and treatment services.

### 1.2. Workplace Flexibility Helps

Workplace flexibility refers to the ability of employees to have control over the timing, duration, and location of work [17]. Flexible work arrangements (FWA) are often promoted within organizations as a means to support employees in meeting their work, family, and personal responsibilities and as a way to achieve work–life integration [17,18]. Despite increased attention to workplace flexibility, employees and employers often report mixed experiences that have been attributed to variations in how workplace flexibility is implemented and viewed [19]. These experiences are the result of a variety of factors, from type of flexibility offered (telework, flextime, part-time, or leaves of absence, along with position within the organization and job autonomy), workplace culture and climate, and the variation found in the implementation of formal policies across organizations [20,21,22]. For organizations with a commitment to diversity and inclusion, the ability to access and use flexibility is an indicator of the degree to which the organization is inclusive [23,24].

Employees have two possible mechanisms to request flexibility: informal and formal. Informal flexibility is achieved through negotiations with co-workers or supervisors to make temporary adjustments to schedules to meet employee personal or family needs [25,26]. Formal adjustments are accessed most frequently through HR staff and are pursued when problems are serious or persistent [27]. In organizations with no clear policies addressing how to successfully implement and maintain flexibility, both informal and formal adjustments can be challenging for employees and managers [28,29]. Employees are sometimes fearful of using flexibility policies because of career backlash, and possible job loss, from flexibility stigma [30,31].

The need for informal and formal workplace flexibility is particularly acute for family members providing disability care who need to maintain employment, as revealed through focus group studies and interviews [32,33]. Because of the nature of disability care and the absence of community supports, requests for flexible work arrangements can be frequent and crisis-driven, and can involve long absences from work [34].

### 1.3. HR Professionals as Key Influencers

HR professionals are often directly responsible for the design and oversight of diversity and inclusion programs, employee assistance programs, benefits, business strategy and leadership, compensation, and metrics. In many cases they may also participate as key influencers of organizational climate and culture through their functional relationships with top executives and managers [27]. Through their collaborative efforts, HR professionals can help to create a health-promotive workplace culture in which exceptional care responsibilities are recognized by all members of the organization [35]. However, an international survey of HR professionals reported that most organizations still view requests of this nature as the enactment of a special benefit that can have high organizational costs, and 79% considered flexibility requests for disability care only on a case-by-case basis [36].

Three elements within organizations are posited to influence HR professionals’ flexibility-request decisions: the presence of *formal policies permitting FWA*, the perceived *strength of the business case for flexibility*, and the *workplace culture*. Formal FWA are written into organizational policy and require HR approval. Studies of U.S. employers found that 81% allow at least some classes of employees’ flexible arrival and departure hours at work, although only 59% permitted all workers’ flextime, which suggests that flexibility is often dependent on job type within an organization [37].

Through her analysis of data from an international survey of HR professionals, Huffstutter [35] found that *belief in the business case* was among the strongest predictors of the likelihood to grant flexibility requests. Flexibility has been associated with improved recruitment, reduced absenteeism and turnover, worker engagement, increased productivity and financial performance, and better client service [29,38,39].

The workplace culture of an organization consists in the assumptions, beliefs, and values held in common by employees regarding the extent to which their organization should support the work–family fit of its members [40]. Despite the presence of formal policies, employees may not feel free to access flexible work arrangements if they expect a negative reaction by co-workers and supervisors who expect high levels of uninterrupted attendance, and who may feel special arrangements are unfair and that flexible work arrangements are not an accepted part of the workplace culture [41,42].

### 1.4. Diversity Training: What Works?

Diversity training approaches, while varied, typically include increasing awareness and skill building [43,44]. A recent study by Dimoff and Kelloway [45] demonstrated the effectiveness of training organizational leaders to improve their awareness of, and response to, employees’ mental health concerns. Awareness refers to declarative knowledge regarding what is known about a particular social identity group. Behavioral learning occurs when trainees are provided information on desired actions that stem from having increased awareness. Knowledge of the organization’s belief in diversity, commitment to diversity goals, and expected standards of behavior are thought to influence attitudes and behaviors of trainees [46]. Action planning refers to the behavioral intentions that the trainee exhibits as a result of experiencing changes in awareness and attitudes, and signals the trainees’ ability to carry the training back to their jobs [47]. Skill building can result when these intentions are carried out in practice through interactive exercises, during which HR professionals react to real or simulated employee dilemmas. A meta-analysis of 260 independent samples of diversity training studies finds that the most effective types of diversity training programs are those that: are designed to increase both diversity awareness and skills, occur across multiple sessions, and use a variety of learning techniques (on-line, in person, group work; [43] (p. 15)). To attain full inclusion and respect for diversity, ongoing training should result in an environment “in which all employees feel valued, respected and supported” [48] (p. 51). The diversity training program that was developed for this study involved both on-line didactic sessions and breakout group exercises, and required working through case examples that challenged them to put awareness and skills into practice plans relevant to the staff that they would be advising.

## 2. Theory and Hypotheses

Social cognitive theory (SCT; [49,50]) posits individuals translate knowledge into behavior through a process of acquiring *knowledge*, enhancing perceived *self-efficacy*, weighing outcome *expectations* regarding costs and benefits for different behaviors, and analyzing perceived *facilitators* and social and structural *impediments* to the behavior change [50]. Individuals are more likely to apply newly learned behavior if they believe they can produce desired effects by their actions; this belief is known as *self-efficacy* [50]. Research suggests that the employees’ ability to transfer diversity training back to their jobs partially depends on their own self-efficacy [51]. Using scripts in diversity training that represent a model that trainees can follow provides mastery experiences [52], potentially heightening self-efficacy. Acceptance and building of positive attitudes can also result from exposure to positive narratives about people with different life experiences [53]. Additionally, the context in which the training takes place can be a significant facilitator of motivation to learn and apply training outcomes. However, an absence of belief in the business case for diversity among decision-makers and the lack of an inclusive workplace culture can act as structural impediments to learning [54]. Diversity training embedded in a larger program of diversity initiatives in a workplace, including managerial commitment to inclusion and the formation of supportive networks, has been found to lead to more significant changes in knowledge and more lasting behavioral outcomes [43].

Using our knowledge of the challenges faced by employed parents giving disability care, we partnered with the U.S. subdivision of a large multinational organization that provides audit, tax, and advisory services. This partnership was a key element of the organization’s efforts to develop a Disabilities Network to “educate and increase awareness, support career development, influence policy decisions, and participate in community activities” [55] (p. 21). This partnership included an initiative to develop and test a group-specific (exceptional caregivers) diversity training program aimed at raising HR professionals’ confidence in their ability to provide support to their employees raising children and youth with disabilities. The organization joined this effort as part of its commitment to diversity and inclusion.

Using propositions found within SCT, along with evidence from the HR training, diversity and disability, work, and care literatures, we proposed three related hypotheses:

**Hypothesis** **1.**Participation in a group-specific, combined (awareness and behavior-based), two session diversity training program increases *HR knowledge about disability care*, and bolsters *self-efficacy* and *likelihood to carry out supportive HR practices*.

**Hypothesis** **2a.***Likelihood to grant workplace flexibility* after training will be predicted by *knowledge of disability care, HR self-efficacy, prior disability awareness training, perceptions of a positive work–life culture*, and *belief in the business case for flexibility*.

**Hypothesis** **2b.***Likelihood to grant workplace flexibility after training* is moderated by the relationship between *knowledge of disability care* and *the belief in the business case for flexibility*, such that the relationship is stronger for those who gained more knowledge from the training and reported greater belief in the business case for flexibility.

**Hypothesis** **3.**HR characteristics including *length of time in current position* and *within the organization* will predict *knowledge and perceived efficacy in making FWA decisions*.

## 3. Methods

### 3.1. Participants

Through the efforts of the Disability Networks of the organization, 90 HR professionals employed by the organization at multiple sites in the US were invited to participate in the training. A total of 64 (71.1%) enrolled and completed both intervention training sessions and associated instruments (See Table 1), a sufficient sample given the total number of HR professionals working in the US branch of the corporation [56,57]. The majority identified as White, Non-Hispanic (73%), with 11% African American, 8% Hispanic/Latino, 6% Asian, and 2% Native American/Pacific Islander/Alaskan Native in the sample. Most participants were female, had a four-year college degree, did not hold additional professional certifications, and had less than 11 years of HR experience. About half had prior corporate training in disability or diversity.

### 3.2. Procedure

This targeted training intervention was designed to bridge the gap between the workplace needs of employed parents of children and youth with disabilities and the business objectives of organizations. The training content and supporting materials were developed based on prior research with families of children with disabilities, workplace supervisors, and HR professionals [32,33]. The training consisted of two sessions, each lasting approximately two hours and scheduled two weeks apart, delivered through an interactive online training platform used by the collaborating organization. The online training modality was chosen since it provided the opportunity for HR participants from geographically scattered subdivisions of the company to come together for a joint training experience. Three weeks before the first session, HR managers in the U.S. received an email from a corporate leader inviting them to participate in trainings. Those managers who were interested clicked on a link that took them to an informed consent statement. If they agreed to participate, they generated a confidential code, then were directed to an online pretest. After completing the pretest, they were sent a link to the online training manual [58]. After completing both training sessions, they received an email asking them to enter their confidential code, and then were sent a link to the online post-test.

Both synchronous online training sessions were conducted by two trainers who provided content and guided exercises to participants through slides and live audio and video feeds, supplemented by materials in the accompanying training manual. The e-learning platform allowed participants to ask clarifying questions in real time using audio connections and to interact in small groups for exercises through audio conferencing.

The first session provided participants with a broad understanding of disabilities affecting children, difficulties associated with disability care, laws and policies protecting the rights of children with disabilities and their families, and challenges faced by parents managing both employment and exceptional care. Towards the end of the session, participants divided into smaller on-line groups to work through a case study using session content. Using case studies or simulated experiences has been shown to increase efficacy among trainees [43].

Concepts presented in Session 2 related to challenges faced by HR professionals when presented with requests for flexibility due to complex family demands. Work-based solutions supported by earlier research with human resource professionals were presented. Participant questions were encouraged throughout the session, and the training stopped at points at which further clarification was requested. A pre-recorded case study was presented, and breakout groups of participants discussed dilemmas facing the HR manager working through flexibility requests of an employee giving exceptional care, and their own related experiences.

The impact of the training on HR knowledge and self-efficacy was assessed using a design with *non-equivalent dependent variables* (NEDV) [59,60] to enhance internal validity. Participants’ pretest and posttest scores were assessed through bivariate analyses that measured differences in scores on the trained (knowledge of dependent care, disability care, community resources, HR self-efficacy) and untrained items that were conceptually relevant NEDVs (belief in business case, health promotive workplace culture). The same contextual threats to internal validity, such as historical events or administrative policy shifts in the organization, could operate on both sets of variables. Multiple regression analyses were also conducted to determine the relative impact of the training and HR professionals’ characteristics on the outcome variables of interest (likelihood to grant workplace flexibility for physical health, for mental health, and for childcare). Interaction terms were entered into the main effects model following the procedure outlined by Aiken and West [61].

### 3.3. Measures

The first two measures used in the study were developed as self-ratings of knowledge of two specific domains relevant to HR decisions on granting employee flexibility: dependent care and community resources. The items had been reviewed by content-area experts, and analyzed for reliability and validity as part of a study of workplace flexibility with 551 HR professionals as respondents [35,62].

*Perceived knowledge of dependent care* was measured through 9 items collecting participants’ self-ratings of their level of knowledge relating to care of dependents using a scale from 1 (*very little or almost no knowledge*) to 5 (*very knowledgeable*). Sample items included: “Adolescent/young adult development, 13–25 years” and “Children’s mental health concerns and treatment”. Responses were summed and averaged to create the Knowledge of Dependent Care Scale (range 1–5; α = 0.92).

*Perceived familiarity with community resources* was measured through participant ratings of their perceived level of familiarity with 11 community resources by selecting a number ranging from 1 (*very unfamiliar*) to 5 (*very familiar*). Sample resource items were “Caregiver support groups,” and “Disability resources”. The items were summed and averaged to create an acceptably reliable Familiarity with Community Resources Scale (Range 1–5; α = 0.95).

*Knowledge of disability care* used 10 multiple-choice items to assess participants’ mastery of knowledge presented in the trainings. For example, a question asked the most common reason that Family and Medical Leave (FMLA) was not used by employees raising children with special health care needs, having participants select 1 of 5 possible answers. Items were coded so that if the participant’s response on the multiple-choice question was correct, a score of 1 was assigned to that item, and if incorrect 0. Once all items were coded for correct responses, they were summed to create the Knowledge of Disability Care Index (range 0–10).

*HR perceived self-efficacy* was assessed through 13 items that were specifically developed for this study using a procedure developed by Bandura [63]. Participants were asked to rate their level of confidence in carrying out 13 inclusion practices using a scale that ranged from 0 (*very little confidence*) to 100 (*quite a lot of confidence*). For example, trainees rated their self-efficacy to “Calculate approximately how many employees in your organization have children of any age with disabilities”. Items were summed and averaged to create the HR Self Efficacy Scale (range 0–100; α = 0.95).

*Belief in business case for flexibility* was measured through 15 items supplying organizational reasons for granting flexible work arrangements such as “improves employee retention” and “decreases employee absenteeism”. Participants indicated their level of agreement with each reason using a 5-point Likert scale ranging from 1 (*very weak*) to 5 (*very strong*). Responses to all 15 items were summed, then averaged to compute the Business Case for Flexibility scale score (range 1–5; α = 0.94). In a prior study with 555 HR participants, a similarly high level of reliability was obtained (α = 0.95; [35]).

*Workplace culture* of participants was assessed using 4 items from the Work-Family Culture Scale developed by the Families and Work Institute [64] combined with five items from the Health Promotive Workplace Culture Scale [35]. Items rated included: “There is an unwritten rule at my place of employment that you can’t take care of family needs on company time”. Participants indicated their agreement with each item using a 5-point Likert scale ranging from 1 (*strongly disagree*) to 5 (*strongly agree*). Items were summed and averaged (range 1–5; α = 0.69). In prior studies, both the Workplace Culture Scale ([64]; α = 0.74), and the Health Promotive Workplace Culture Scale [35]; α = 0.70) had acceptable reliability.

*Likelihood to grant flexible work arrangements* was assessed through items developed and tested on a national sample of HR professionals [35]. Participants were asked to rate on a scale of 1 (*very unlikely to approve request*) to 5 (*very likely to approve request*) the likelihood that a flexible work arrangement would be approved within their organization based on the reason indicated. Items were summed and averaged to create the Likelihood to Grant Flexible Work Arrangements Scale (range 1–5; α = 0.91). Three sub-scales measured Likelihood to Grant Flexible Arrangements for Physical Health (8 items, e.g., “employee caring for family member with terminal illness”), Mental Health (4 items, e.g., “employee’s child has ongoing mental health therapy”) and Childcare Reasons (4 items, e.g., “employee has short-term child care difficulties”).

## 4. Results

**Hypothesis 1** was assessed through *t*-tests for both trained (knowledge of human development, knowledge of disabilities, familiarity with community resources, HR self-efficacy, likelihood to grant flexibility) and untrained items (workplace culture, belief in business case) and their associated effect sizes ([65]; see [66] for procedure). Participants’ scores significantly increased from the pretest to posttest on the trained items: knowledge of dependent care, *t* (64) = 5.96, *p * < 0.001, *d_z_* = 0.74; knowledge of disability care, *t* (64) = 12.60, *p* < 0.001, *d_z_* = 1.57; familiarity with community resources, *t* (64) = 2.52, *p* < 0.05, *d_z_* = 0.0.31; and, HR self-efficacy, *t* (64) = 8.81, *p* < 0.001, *d_z_* = 1.10. Analysis of the planned behavior items (likelihood of granting flexibility for physical health, mental health, and child care reasons) significantly increased for child-related care reasons only (*t* (64) = 2.05, *p* < 0.05, *d_z_* = 0.23). In contrast, the change in scores for untrained items, the business case and workplace culture (NEDV), did not reach significance.

**Hypotheses 2a, 2b**. Table 2 shows the intercorrelations of the variables for Hypotheses 2a and 2b. Five of seven of the variables in our study had significant associations with likelihood to grant flexibility for dependent care. Of the five, three of the variables had positive, significant, and large or medium associations: workplace culture (*r* = 0.533, *p* < 0.01), belief in the business case (*r* = 0.440, *p* < 0.01), and familiarity with community resources (*r* = 0.320, *p* < 0.01) [67]. A fourth trained variable, self-ratings of their knowledge of dependent care (*r* = 0.265, *p* < 0.05), revealed a small and positive correlation. Surprisingly, knowledge of disability care produced the only non-significant association with likelihood to grant flexible work arrangements.

Table 3 presents the results of the hierarchical regression models predicting likelihood to grant flexibility for physical health, mental health, and child care reasons (Hypothesis 2a and 2b). Knowledge of human development and knowledge of disabilities did not reach statistical significance in any of the main effects models. The final interaction model for likelihood of granting flexibility for physical health reasons accounted for 24% of the variance, with one interaction term uniquely contributing 7% of the variance to the overall model. The interaction term of belief in the business case and knowledge of disabilities indicated that, above and beyond, the main effect association was of belief in the business case; HR professionals who believed more strongly in the business case and who scored higher on the knowledge of disability items at posttest were more likely to grant flexibility for physical health reasons.

The regression model for likelihood of granting flexibility for mental health reasons had three significant main effects and one significant interaction, accounting for 22% of the variance in the model. Prior disability awareness training was the strongest predictor, followed by the business case for flexibility then workplace culture. The interaction term model produced a significant effect, suggesting that those participants who endorsed the business case for flexibility and who gained knowledge of disability were more likely to grant flexibility for mental health reasons.

The final interaction model accounted for 41% of the variance in likelihood of granting flexibility for childcare reasons, with one significant interaction term. While the interaction of belief in the business case x knowledge of disabilities positively predicted likelihood of granting flexibility for childcare reasons, the interaction of workplace culture x knowledge of disability was negatively associated. This finding suggests that those who felt that the workplace culture was more family friendly and who received higher knowledge scores in the disability training were less likely to grant flexibility for childcare reasons.

**Hypothesis 3.** The results of the simultaneous regression models assessing whether HR characteristics predicted knowledge of dependent care, knowledge of disability care, and self-efficacy after the training are presented in Table 4 (Hypothesis 3). The predictors accounted for 43% of the variance of knowledge of dependent care (*F* (7, 46) = 6.67, *p* < 0.001). Familiarity with community resources (*β* = 0.52, *p* < 0.001) and self-rated knowledge of dependent care prior to the training (*β* = 0.32, *p* < 0.01) positively and significantly predicted self-rated knowledge of dependent care after the training.

The predictors explained 14% of the variance in knowledge of disabilities (*F* (7, 46) = 2.02, *p* < 0.05). Knowledge of disabilities prior to the training was the only significant predictor of knowledge of disabilities after the training (*β* = 0.56, *p* < 0.001).

Three predictors accounted for 41% of the variance in self-efficacy, *F* (7, 46) = 6.15, *p* < 0.001). The most significant contribution made to the prediction of self-efficacy after training was length of time in current job (*β* = −0.59, *p* < 0.001), followed by length of time in HR field (*β* = −0.41, *p* < 0.01). Less experience was associated with higher self-efficacy ratings at Time 2. Knowledge of dependent care prior to the training (*β* = 0.21, *p* < 0.05) also significantly predicted self-efficacy.

## 5. Discussion

Human resource professionals participating in targeted diversity training made significant gains in their knowledge about exceptional caregiving that employed parents provide to their children and youth with disabilities. Participants also showed significant increases in self-efficacy, specifically their reported confidence in taking workplace actions to support employees with exceptional caregiving responsibilities. Prior research found that employees providing exceptional care are not typically within groups that HR professionals usually categorize as needing support to better manage the work–family interface [27]. Trainings that increase awareness of employee diversity of caregiving demands, coupled with improved self-efficacy, likely better equip HR staff to promote workplace inclusiveness.

Flexible work arrangements are one of the specific strategies accessed by employed parents to meet the needs of their children with disabilities, due to decreasing conflict between work and family demands [68]. In the current study, the perceived likelihood of participants granting flexible work arrangements was associated with knowledge of dependent care, familiarity with community resources, belief in the strength of the business case for flexibility, and the perceived support level of workplace culture. Huffstutter [35] reported similar findings in her study of 551 members of an HR professional organization. Human resource professionals who indicated a higher likelihood of granting flexible work arrangements for dependent care also gave higher endorsements to the business case for FWA, reported working in an organization with a supportive workplace culture, and illustrated greater knowledge of dependent care issues. Moreover, both workplace culture and employee self-efficacy have been identified as key variables in the transfer of training within organizations [69].

The findings address our three research questions that were based partly on the theoretical propositions of SCT [49], confirming that HR professionals are more likely to apply new knowledge and behavior if they perceive a direct benefit, believe that they can master the new behavior, and are supported by facilitators within the organization who endorse the change [43]. Our results align with those of existing diversity literature arguing that varied training modalities, including use of scripts to formulate action plans, enhance learning and planned actions [70].

Finally, some participant characteristics predicted study outcomes. Those who are newer to the HR field or their current job may be more highly motivated to master the material that is presented in a training format [71], and rated their own self-efficacy higher after the training than those with more experience. The accumulation of HR experience of participants was inversely associated with their confidence in their ability to support employees engaged in exceptional care. It is probable that HR employees who were in their current positions for a shorter time period were more recently hired or promoted, a factor that could have contributed to higher ratings of self-efficacy.

### 5.1. Implications for Workforce Development on Dependent Care Diversity

Targeted training through a combined (awareness and behavior-based) two session diversity training program about dependent care diversity can provide opportunities for HR professionals to become more knowledgeable and skillful in developing actionable goals related to employees with exceptional care experiences. Focus group research has reported that both employed exceptional caregivers and HR professionals alike have an array of concerns when discussing issues such as equity, disclosure, resource access, management of confidential information, and stigmatization in the workplace [27]. These concerns highlight the need to make the corporate culture itself more inclusive of workers who experience either disabilities or engage in disability care [15]. Effective training can address concerns, offer practical strategies, and bolster self-efficacy for managing these sensitive and critical workplace exchanges [3].

Key for the employed parents of focus in this research is a clear organizational pathway to workplace flexibility [72]. Requests for FWA often require disclosing the reason or need for flexibility. For employees with invisible differences, disclosing their status to gain access to a benefit, or for protection from discrimination, presents a complicated dilemma [24]. Disclosing information about a child’s disability, especially a child’s mental health diagnosis, may bring on stigmatization, closer scrutiny, and judgment from supervisors and coworkers [73]. Adding FWA policies and practices as a dimension of the inclusive workplace culture necessitates training initiatives that help staff acquire or increase their range of knowledge and skills about workplace flexibility, as well as awareness and prevention of stigmatization of employees who make requests due to disabilities of their family members, or their own disabilities [24].

### 5.2. Limitations

Participants were a convenience sample drawn from a single organization that largely employs auditors, tax advisors, and business professionals to provide consultation and services to other companies. Although the sample was national in scope, it was relatively small, with only 64 participants. All the measures used in the study were self-reports and knowledge tests completed by the HR trainees, raising the possibility that our results were affected by common method bias (CMB; [74,75]). This issue was, at least, partially addressed by the incorporation of established measures with different scale types using varied anchor labels [76]. Statistical methods of determining and correcting for common method variance (CMV; [77]) were not used in this study because of the small sample size.

The training intervention study also had limited controls for internal validity, since random assignment to intervention and control groups was not possible given practical constraints. The inclusive workplace culture at the organization may have affected the outcomes of the training. There is also the possibility that participants’ survey responses were affected by the substantial initiative in the organization on improving the inclusion of those employees and customers who experienced their own disabilities or cared for family members with disabilities.

Also, actual behaviors of HR professionals as they encountered exceptional caregivers were not tracked after the conclusion of training. Nor was the impact of training the organization’s human resource professionals on employee outcomes assessed. Incorporating criterion variables with ratings of HR behavior and employee outcomes given by observers, rather than trainees, could help to mitigate CMB in future studies [75].

### 5.3. Future Research Directions

To move inclusion in the workplace forward, subsequent investigations with larger samples and more highly controlled experimental designs would be desirable. These studies might also provide post-training coaching and mentoring to promote greater self-efficacy and utilization of inclusive practices [78] and track participant behavior over time after training is completed. Studies are required to determine whether increasing knowledge and supportive behaviors leads to changed actions for managers and supervisors addressing the needs of parents of children with disabilities and positive employee outcomes [79]. Research that examines line manager decision-making regarding flexibility requests suggests that they may customize individual flexibility requests when workers proactively take initiative [80,81]. Follow-up studies of employee outcomes that track the trajectory of workplace engagement of exceptional caregivers and their perceived levels of inclusion are needed. Studies could also examine the relationship between HR efforts to include workers with exceptional care responsibilities and the access of these workers to opportunities to engage in job crafting that would permit them to develop more appropriate work arrangements [80]. Last, efforts to understand how targeted diversity training programs, such as this one, fit within larger organizational health practices are needed to see how they might fit with organizational culture change efforts.

Finally, cross-national studies of management approaches to supporting employees who provide exceptional care might be very fruitful [82,83]. National policy supports for families raising children with disabilities vary widely, in terms of income supports, access to specialized healthcare and educational services, and workplace flexibility [6]. When employing companies operate in very different national policy environments, those designing training for management staff in multinational organizations will need to draw on best practices within their national policy and cultural contexts.

## Figures and Tables

**Table 1 healthcare-10-02361-t001:** Demographic characteristics of participants (N = 64).

Characteristic	*N*	*%*
Gender		
Female	52	81.3
Male	12	18.8
Race/Ethnicity		
White NH	47	73.4
African American NH	7	10.9
Hispanic/Latino	5	7.8
Asian	4	6.3
Native American/Pacific Islander/Alaskan Native	1	1.6
Education		
Graduate degree	14	21.9
College	43	67.2
Some college	5	7.8
Has other certification		
SPHR	10	15.7
PHR	13	20.4
CPA	4	6.3
Special education teacher	1	1.6
Community relations	1	1.6
Prior training		
FWA	12	18.8
Work-life	9	14.1
FMLA	5	9.1
ADA	11	17.2
Disability	34	53.1
Diversity	30	46.9
	*M*	*SD*
Years in current job	4.35	4.10
Years in HR	10.53	6.35

**Table 2 healthcare-10-02361-t002:** Means, standard deviations, and correlations for study variables (*N* = 64).

	Variable	1	2	3	4	5	6	7	8	9	10	11	12	13	14	15	16
1	Business case T1	1															
2	Business case T2	0.55 ***	1														
3	Fam com resources T1	0.21	0.14	1													
4	Fam com resources T2	0.17	0.30 *	0.35 **	1												
5	Workplace culture T1	0.33 ***	0.24	0.14	0.14	1											
6	Workplace culture T2	0.43 ***	0.27 *	0.06	0.17	0.75 ***	1										
7	Knowledge dep care T1	0.05	−0.09	0.54 ***	−0.00	0.10	0.09	1									
8	Knowledge dep care T2	0.17	0.17	0.53 ***	0.56 ***	0.27 *	0.28 *	0.33 **	1								
9	Knowledge disability T1	−0.02	−0.12	0.10	−0.00	0.05	−0.14	0.32 *	0.22	1							
10	Knowledge disability T2	0.04	0.03	0.49 ***	−0.02	0.10	−0.03	0.10	0.09	0.47 ***	1						
11	Self-efficacy T1	0.08	−0.12	0.33 **	0.26 *	0.08	0.11	0.59 ***	0.37 **	−0.06	−0.06	1					
12	Self-efficacy T1	0.09	0.14	0.14	0.46 ***	0.15	0.23	0.21	0.50 ***	0.06	0.09	0.30 *	1				
13	Likelihood grant	0.41 ***	0.39 ***	0.10	0.46 **	0.49 ***	0.51 ***	0.17	0.00	0.18	−0.03	−0.01	0.22	1			
14	flex child care Likelihood grant flex mental health	0.38 ***	0.39	0.16	0.24	0.38 ***	0.38 ***	0.05	0.04	0.19	0.04	0.02 *	0.28 *	0.78 ***	1		
15	Likelihood grant	0.40 ***	0.38 ***	0.28 *	0.18	0.39 ***	0.42 ***	0.19	-0.07	0.26 *	−0.07	0.15	0.32 *	0.76 ***	0.84 ***	1	
	flex phys health																
16	Total likelihood	0.44 ***	0.44 ***	0.15	0.32 **	0.50 ***	0.53 ***	0.03	0.26 *	0.00	−0.05		0.03	0.30 *	0.93 ***	0.88 ***	0.84 ***
M		4.08	4.20	3.20	3.49	3.67	3.77	2.92	3.64	4.36	6.93	50.67	75.27	4.07	4.26	4.39	3.98
SD		0.56	0.56	0.88	0.69	0.59	0.53	0.92	0.70	1.66	1.56	21.79	14.91	0.60	0.52	0.42	0.48

Note. * *p* < 0.05; ** *p* < 0.01; *** *p* < 0.001 Fam com resources = Familiarity with community resources; Knowledge dep care = Knowledge of dependent care; Knowledge dis = Knowledge of disabilities; Likelihood grant flex = Likelihood to grant flexibility.

**Table 3 healthcare-10-02361-t003:** Hierarchical multiple regression models of likelihood of granting flexibility for physical health (*n* = 60), mental health (*n* = 60), and child care reasons (*n* = 62).

			Step 1		Step 2	
R^2^	Outcome	Predictor	Β	Sr^2^	β	Sr^2^
0.25 **	Physical health	KHD	−0.04	--	−0.04	--
		KD	−0.55	--	0.06	--
		KCR	−0.18	--	−0.14	--
		SE	0.06	--	0.02	--
		DA	0.08	--	0.13	--
		BC	0.30 *	0.08	0.31 *	0.12
		WPC	0.36 **	0.11	0.33 **	0.14
		BC x KD	--	--	0.26 *	0.07
		WPC x KD	--	--	0.00	--
0.22 **	Mental health	KHD	−0.06	--	−0.09	--
		KD	−0.07	--	0.06	--
		KCR	−0.00	--	0.04	--
		SE	0.11	--	0.08	--
		DA	0.13	--	0.18	--
		BC	0.29 *	0.08	0.30 *	0.09
		WPC	0.30 *	0.09	0.27 *	0.07
		BC x KD	--	--	0.30 *	0.07
		WPC x KD	--	--	0.11	--
0.41 ***	Childcare					
		KHD	−0.05	--	−0.01	--
		KD	−0.07	--	0.06	--
		KCR	0.07	--	0.11	--
		SE	0.07	--	0.03	--
		DA	0.23 *	0.05	0.27 *	0.06
		BC	0.27 *	0.06	0.29 **	0.08
		WPC	0.44 ***	00.18	0.43 ***	0.17
		BC x KD	--	--	0.21 *	0.04
		WPC x KD	--	--	−0.21 *	0.04

Note: KHD = knowledge of human development; KD = knowledge of disabilities; KCR = knowledge of community resources; SE = perceived self-efficacy; BC = belief in the business case for flexibility; WPC = workplace culture; Sr^2^ = semipartial squared correlation. * *p* < 0.05. ** *p* < 0.01. *** *p* < 0.001.

**Table 4 healthcare-10-02361-t004:** Simultaneous regression models of HR characteristics predicting training outcomes.

Characteristic	Knowledge of Dependent Care		Knowledge of Disabilities		HR Self-Efficacy	
	β	SE	β	SE	β	SE
Length of time current job	−0.18	0.03	−0.08	0.10	−0.59 ***	0.65
Length of time HR	0.13	0.01	−0.05	0.04	−0.41 **	0.26
Knowledge of dependent care T1	0.32 **	0.08	−0.07	0.23	0.21 *	1.55
Knowledge of disabilities T1	0.08	0.05	0.56 ***	0.14	0.11	0.90
Familiarity with community resources	0.52 ***	0.12	−0.10	0.34	0.21 ^T^	2.33
Workplace culture	0.13	0.15	0.04	0.41	0.14	2.83
Business case for flexibility	0.01	0.15	0.10	0.40	−0.02	2.76
*R* ^2^	0.43		0.14		0.41	
*F*	6.67 ***		2.20 *		6.15 ***	

Note: ^T^ *p* < 0.10 * *p* < 0.05 ** *p* < 0.01 *** *p* < 0.00.

## Data Availability

Data from this study are available from the corresponding author upon request.

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
