# Peer review of "Expanding Workplace Inclusion of Employees Who Are Parents of Children with Disabilities through Diversity Training"

_healthcare, 2022, doi:10.3390/healthcare10122361_

Round 1
Reviewer 1 Report
I have appreciated the opportunity to review “Can Training Human Resource Professionals Increase Knowledge and Efficacy Regarding the Needs of Employees who are Parents of Children with Disabilities?”. It examines an important yet less explored issue. Feedback was collected from 551 HR professionals who attended an online training. Still, there are issues in the manuscript that authors may give more attention. To further strengthen the manuscript, I have the following questions and comments for your consideration:
As this paper is about evaluating an intervention, I would like to see the authors to give more attention to the literature on relevant interventions. Are the training content and methods similar to / different from these practices?
A closer connection between the literature review and the training offered is necessary.
The authors state that “With appropriate supports and community resources, parents raising young people with special needs can thrive” (p.2, line 62). Readers may want to know where the supports come from (Employers? Social service providers?).
It is said that diversity training should better be “longer in duration” (p.3, line 137). How long is appropriate? Are two sessions (four hours in total) long enough?
It would be better to give further justification for the three proposed hypotheses. It would be especially helpful to show the research gaps, and use the discussion of these gaps to introduce the hypotheses.
More details about the training course should be helpful.
Why the online mode was adopted? Was it due to the pandemic?
It may be good to briefly discuss the ways how the authors addressed possible social desirability.
The authors may consider discussing what organizational changes may be needed for enterprises. In addition, it may also be nice to mention how interventions can be provided by non-profit organizations to support these changes. The authors may make reference to Kwan’s (2022) work on mezzo–level interventions.
Thank you.
Author Response
Dear Reviewer 1:
Thank you for the opportunity to respond to your suggestions. We hope that you will agree that the changes greatly strengthen the submission. Please see response to Reviewer 1 in the attached table.

Reviewer 2 Report
This is an excellence piece of work which deals with the effects of targeted diversity training on human resource (HR) professionals’ knowledge of work-family experiences of these parents, and on their self-efficacy in providing workplace supports. Its is an innovative issue of discussion as many articles focus on younger ages of people with disabilities, so research on this target group is quite limited.
I would suggest you add some references about "Τhe Cultural Construction of the Different Other: Expanding Foucault’s Theory". If you google it you can find the article and I believe that it would enrich your manuscript. Foucault's contribution in deep theoretical analysis on diversity is exceptional. You can also add something about how ready the workplace is to accept diversity.
The only recommendations for improvement I have to add deal with the methodology chapter. It would be interesting if you add some more information about the method you chose to follow and why as well as for the research tool. I didn't see any research questions also that are useful for your conclusions too. In the Conclusion you can answer directly those research questions.
I wish you all the best and after those amendments I will be more than happy to suggest the acceptance of this paper.
Author Response
Dear Reviewer 2:
Thank you for your thoughtful review of our manuscript. We appreciate the feedback. You can see our individual responses to your comments in the attached document under Response to Reviewer 2.

Reviewer 3 Report
Dear authors,
I enjoyed reading your article. The topic is interesting, and there is clearly potential value in this line of research to academics and practitioners.
Major comments:
Title:
1) While I find the topic itself intriguing, I would think first about changing the title of the article. The current title is - "Can Training... Increase Knowledge and Efficacy...". Of course training can do that. This is not at all interesting for readership. Surely, whatever you train for is likely to lead to knowledge and efficacy on the subject, at least temporarily so.
I would consider including in the title 'workplace flexibility', 'business case for flexibility', 'organizational culture' (and so on), as your hypotheses concern these matters.
Literature:
2) The differences (comparison and contrast) between child disability, spouse disability, parent disability, personal self-disability, etc. are not discussed. I believe if you go into great length discussing those matters properly, this paper can have greater value for the reader. I would also consider referring to combined effects - disabled partner and disabled child, or personal disability and disabled child, or having multiple children with disability.
With all these possibilities, currently, it is unclear why specifically parenting a child with disability worth special attention.
This, I believe, is the most important comment in my review - extending the background is crucial.
3) The structure of the literature review can be improved - the section of 'Workplace Flexibility Helps' provides a lot of background, while as reader I don't understand how this background connects to the previous parts, for instance, of the paper as a whole. Some explanation about the structure of the paper to the reader could help. Similarly with the title 'Diversity Training: What Works?' - the structure is not very intuitive, hence explanation is needed.
Methods:
4) There is only one organization, with 64 self-selected HR professionals (out of 90 employed). This is a weakness of this paper, so I would expect to have an extended context of the organization. At least a full paragraph on the organization, industry, it's diversity metrics, size, etc. Some context is provided in the limitation section, and I would like to see an extension of that in the method section. Why was this organization chosen? Can we expect the results to be generalizable across industries? Maybe it is more relevant to knowledge industries specifically (given the greater likelihood for flextime/flexspace job opportunities).
5) All the above should be reflected in the discussion section.
Minor comments:
1) "employees with dependent care responsibilities" - do you refer to employees with children? please simplify and clearify
2) Informal flexibility - please also refer to the concept of 'job crafting'.
3) "HR professionals are not only directly responsible...... (too wordy and somewhat inaccurate)". Please simplify and shorten to "HR professionals are key influences of organizational climate and culture...".
Also, "through their functional relationship with top executives" - this depends on the type of HR department. I would say "In many cases..." instead of generalizing (namely, "In many cases, HR professionals are key influences of organizational climate and culture...")
4) "through their strategic work" - what do you mean by that?
5) Method - how were HR professionals chosen? was there a reward? Did some of the dropped during the training? Please provide some more information.
5) Measure "Likelihood to grant flexible work arrangements" - you mentioned 3 sub-scales. Please provide sample item in text for each of the sub-scales.
6) Correlation table - The order of variables is not clear. Why the first variable is 'business case'? I would expect training variables to come before the other variables - with the dependent variables first. Could you also mark in the table the training variables? Maybe in italics? underline? Just so it's easier to locate the most important variables.
In sum, I really enjoyed reading your paper. It is well-written, and I wish the authors best of luck with their manuscript.
Author Response
Dear Reviewer 3:
Your thoughtful review of our manuscript was greatly appreciated. We hope that you will agree that the changes made strengthen the paper. Our responses to your suggestions can be found in the attached table under Responses to Reviewer 3

Round 2
Reviewer 3 Report
Dear authors,
I was glad to see that you put effort on addressing previous comments.
Few additional comments to consider:
1. Title - I would suggest making it shorter, more interesting, and as mentioned in previous review, "Does... training affect..." - is not very interesting in my view - since most likely it does affect!
Maybe something along the lines of "Training for inclusiveness: The case of employees who are parents of children with disabilities"
2. "Because a substantial proportion of the workforce cares for at least one child with a disability [12], targeted research is warrented"
Question A - what is a substantial proportion?
Question B - why does a substantial proportion merits a targeted research? It doesn't sound very convincing as a motivation for research in my view
3. "One international study on the global workforce found..." - this paragraph wasn't clear for me. Also, what is "exceptional"?
4. Typo? - "type of flexibility offered (telework, flextime, part-time, or leaves)
5. Discussion - "Our results confirm existing diversity literature" - could you make this a less conclusive statement? I would find it hard that one research can actually confirm anything.
Thank you for your effort
Author Response
Dear Reviewer,
Thank you for your helpful suggestions. We hope that you agree that the changes render the manuscript stronger. Below are our responses to your suggestions:
- Title - I would suggest making it shorter, more interesting, and as mentioned in previous review, "Does... training affect..." - is not very interesting in my view - since most likely it does affect!
Maybe something along the lines of "Training for inclusiveness: The case of employees who are parents of children with disabilities"
Thank you for the suggestion. We have changed the title to reflect your suggestion.
- "Because a substantial proportion of the workforce cares for at least one child with a disability [12], targeted research is warranted"
Question A - what is a substantial proportion?
We have added support for our claim by way of citing the results from the 2019 American Community Survey.
Question B - why does a substantial proportion merits a targeted research? It doesn't sound very convincing as a motivation for research in my view
We removed the sentence.
3. "One international study on the global workforce found..." - this paragraph wasn't clear for me. Also, what is "exceptional"?
We have clarified the sentence.
We removed “exceptional” replacing it with “family care”.
- Typo? - "type of flexibility offered (telework, flextime, part-time, or leaves)
We have added “of absence” to clarify this sentence.
- Discussion - "Our results confirm existing diversity literature" - could you make this a less conclusive statement? I would find it hard that one research can actually confirm anything.
We have softened our statement by removing “confirm” and replacing it with “align with those”.